# Domain Generalization-Aware Uncertainty Introspective Learning for 3D Point Clouds Segmentation

## ABSTRACT

Domain generalization 3D segmentation aims to learn the point clouds with unknown distributions. Feature augmentation has been proven to be effective for domain generalization. However, each point of the 3D segmentation scene contains uncertainty in the target domain, which affects model generalization. This paper proposes the Domain Generalization-Aware Uncertainty Introspective Learning (DGUIL) method, including Potential Uncertainty Modeling (PUM) and Momentum Introspective Learning (MIL), to deal with the point uncertainty in domain shift. Specifically, PUM explores the underlying uncertain point cloud features and generates the different distributions for each point. The PUM enhances the point features over an adaptive range, which provides various information for simulating the distribution of the target domain. Then, MIL is designed to learn generalized feature representation in uncertain distributions. The MIL utilizes uncertainty correlation representation to measure the predicted divergence of knowledge accumulation, which learns to carefully judge and understand divergence through uncertainty introspection loss. Finally, extensive experiments verify the advantages of the proposed method over current state-of-the-art methods. The code will be available.

## KEYWORDS

Domain Generalization, Point Clouds, 3D Semantic Segmentation, Uncertainty Introspective Learning

**ACM Reference Format:**

. 2024. Domain Generalization-Aware Uncertainty Introspective Learning for 3D Point Clouds Segmentation. In *Proceedings of Make sure to enter the correct conference title from your rights confirmation emai (MM'24)*. ACM, New York, NY, USA, 9 pages. https://doi.org/10.1145/nnnnnnn.nnnnnnn

## 1 INTRODUCTION

Point clouds semantic segmentation uses LiDAR to perceive the scene distribution of the three-dimensional world and has a wide range of applications, such as autonomous driving [5, 10, 19], robotics [4, 15] and medicine [35]. Currently, semantic segmentation of point clouds can achieve high accuracy under normal environmental conditions [39]. However, 3D segmentation will inevitably reduce the reliability of environmental perception under adverse conditions. For example, some weather conditions such as fog, snow, and rain will also occur in autonomous driving scenarios. Therefore, improving the generalization of 3D semantic segmentation of point

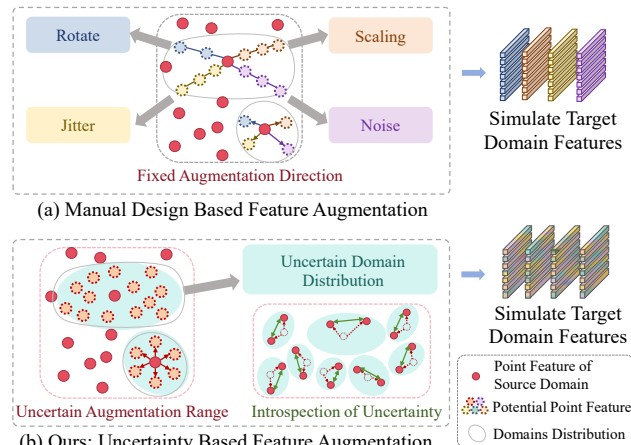

(a) Manual Design Based Feature Augmentation

(b) Ours: Uncertainty Based Feature Augmentation

**Figure 1: Feature augmentation methods essentially aim to simulate the unknown distribution of the target domain during training based on priors. (a) Manually designed priors provide some preset directions for enhancement. (b) The proposed method focuses on the uncertainty of the points and performs uncertain introspective learning, which provides a broad scope for enhancement and can expand the simulated domain distribution.**

clouds under adverse conditions has become an indispensable task with growing significance.

Previous 3D point cloud methods utilize some prior information of the target domain point cloud data to adapt the model to a specified distribution [23, 29, 38]. CoSMix [23] proposes to mix samples from the labeled source domain and the pseudo-labeled target domain to increase the sample space. PolarMix [29] enriches the distribution of point clouds through enhanced strategies of scene-level swapping and instance-level rotation and pasting. SVCN [38] proposes a sparse voxel completion network that assigns semantic labels from the recovered underlying 3D surface in a two-stage manner. Domain adaptation methods can alleviate the domain shift of point cloud segmentation models. Some domain adaptation methods [28, 40] project point cloud data to images to reduce domain shift.

The above methods transfer the model to a distribution using domain-specific data augmentation. However, there are often unseen domains in real-world scenarios, which degrades the generalization and reliability of the model under some extreme conditions. PointDR [31] leverages domain generalization to randomize the geometric style of point clouds and aggregate embeddings to improve model generalization under adverse weather. 3DLabelProp [24] uses past sequences to propagate labels for newly registered scans. Kim *et al.* [11] perform domain enhancement by randomly subsampling point cloud data to simulate the unseen domain. Although feature enhancement through domain random is an effective way, carefully

manually designed feature augmentation has a limited scope, as shown in Figure 1 (a).

Furthermore, the point cloud is essentially a collection of physical points, and each point has uncertainty in the real collection. Failure to take this uncertainty into account will harm generalization performance in unseen target domains. For example, in extreme weather, raindrops and snowflakes will appear at any point in the target domain [31]. Point cloud data under adverse conditions may be missing or occluded, making the model highly uncertain for these samples. Specifically, the data collection process of a point cloud usually uses flight time and target distance information to calculate the three-dimensional coordinates and point cloud characteristics of the target surface [5]. However, the process of collecting spatial structure information usually suffers from sampling bias, which makes the spatial structure information contain great uncertainty. Especially in extreme weather conditions, the active LiDAR pulse system is easily affected by scattering media such as raindrop particles and snow [2]. This will lead to uncertainty such as weakened echo intensity, offset in the measurement distance, and missing points [34]. The uncertainty problem further affects the domain generalization of the point cloud segmentation model.

In this paper, a Domain Generalization-Aware Uncertainty Introspective Learning (DGUIL) framework is proposed for point cloud segmentation. The proposal focuses on the uncertainty of points in domain shift to improve the feature generalization. As shown in Figure 1 (b), our advantage over previous methods is that it provides a broad scope for feature augmentation, which is beneficial to expanding the simulation diversity of unknown distribution target domains. Specifically, Potential Uncertainty Modeling (PUM) is proposed to explore the underlying uncertain point cloud features and generate the different distribution for each point. The PUM enhances the point features over an adaptive range, which provides various information for simulating the distribution of the target domain. Second, the model needs to introspectively learn universally applicable features from the uncertainty distribution. Therefore, Momentum Introspective Learning (MIL) is proposed to learn generalized feature representation in uncertain distributions. The MIL utilizes uncertainty correlation representation to measure the prediction difference of knowledge accumulation. MIL learns to carefully judge and understand divergence through uncertainty introspection loss, which further improves the domain generalization ability.

Our main contributions can be summarised as follows:

- The Domain Generalization-Aware Uncertainty Introspection Learning (DGUIL) framework is proposed for 3D point cloud segmentation, which addresses the point uncertainty in domain shift to improve feature generalization.
- The Potential Uncertainty Modeling (PUM) is proposed to explore the underlying uncertain point cloud features and enhance the point feature with different potential distributions.
- Momentum Introspective Learning (MIL) is proposed to learn generalization features from uncertain distributions with uncertainty introspection loss.
- The performance of this method reaches the current state-of-the-art, verifying the effectiveness of the proposal.

## 2 RELATED WORK

### 2.1 3D Point Clouds Segmentation

3D semantic segmentation aims to assign each point in 3D point cloud data to its corresponding semantic category [3]. Unlike 2D image semantic segmentation [8], which focuses on processing pixel information in images, 3D semantic segmentation needs to deal with the discreteness and irregularity of point clouds [18]. Therefore, 3D segmentation faces more complex spatial modeling and data processing challenges. In recent years, mainstream methods have used deep learning models to learn feature representations in point clouds, which has greatly improved the performance of point clouds. Some methods [12, 32] based on 2D segmentation project 3D point clouds onto images from various viewpoints. However, dimensionality reduction will lose a lot of information. Point-based methods [21, 22] take raw uneven point clouds as input, which require extensive computation. In addition, the voxel-based method [17] divides the three-dimensional space into multiple grids according to a certain scale. The voxel method combined with the recent sparse convolution SparseConvNet [26] can efficiently segment point clouds.

Although a large number of methods have been proposed to improve the accuracy of point cloud semantic segmentation, current methods still have insufficient accuracy when dealing with complex environments. For example, scene segmentation in autonomous driving environments often fails under adverse weather conditions, which poses a great safety risk to the recognition algorithm. Therefore, the proposed method focuses on domain generalization of 3D semantic segmentation to adapt to actual complex scenes.

### 2.2 Generalized Semantic Segmentation

Domain Generalization (DG) refers to generalizing the model to unseen target data from different domains or environments when training the model. The DG model maintains efficient semantic segmentation performance in new domains by learning the robustness of data distribution. Therefore, DG is more challenging than the domain adaptation [6, 13, 14] problem that only needs to adapt to a specific target domain. Generalized 2D semantic segmentation can usually perform style randomization in the input layer, or enhance diverse representations based on visual priors [7]. However, 3D point clouds are unordered and unstructured, with more complex spatial geometry than flat image data [9]. The texture and geometric features of scenes and objects are condensed in the form of point cloud data. Under adverse conditions, the complexity is further exacerbated, making generalized point cloud segmentation more challenging [40]. Domain enhancement-based methods can improve model generalization by mining diverse point cloud features. Kim *et al.* [11] perform domain augmentation by randomly subsampling point cloud data to simulate unseen domains. 3DLabelProp [24] uses past sequences to propagate labels for newly registered scans. PointDR [31] leverages domain generalization to randomize the geometric styles of point clouds and aggregate their embeddings to improve model generalization in adverse weather.

However, point cloud data under adverse conditions have high uncertainties. The transmission of point cloud data may be lost or obscured. Typically in unfavorable weather, rain or snow can block part of the laser, and fog may blur the sample. If the impact

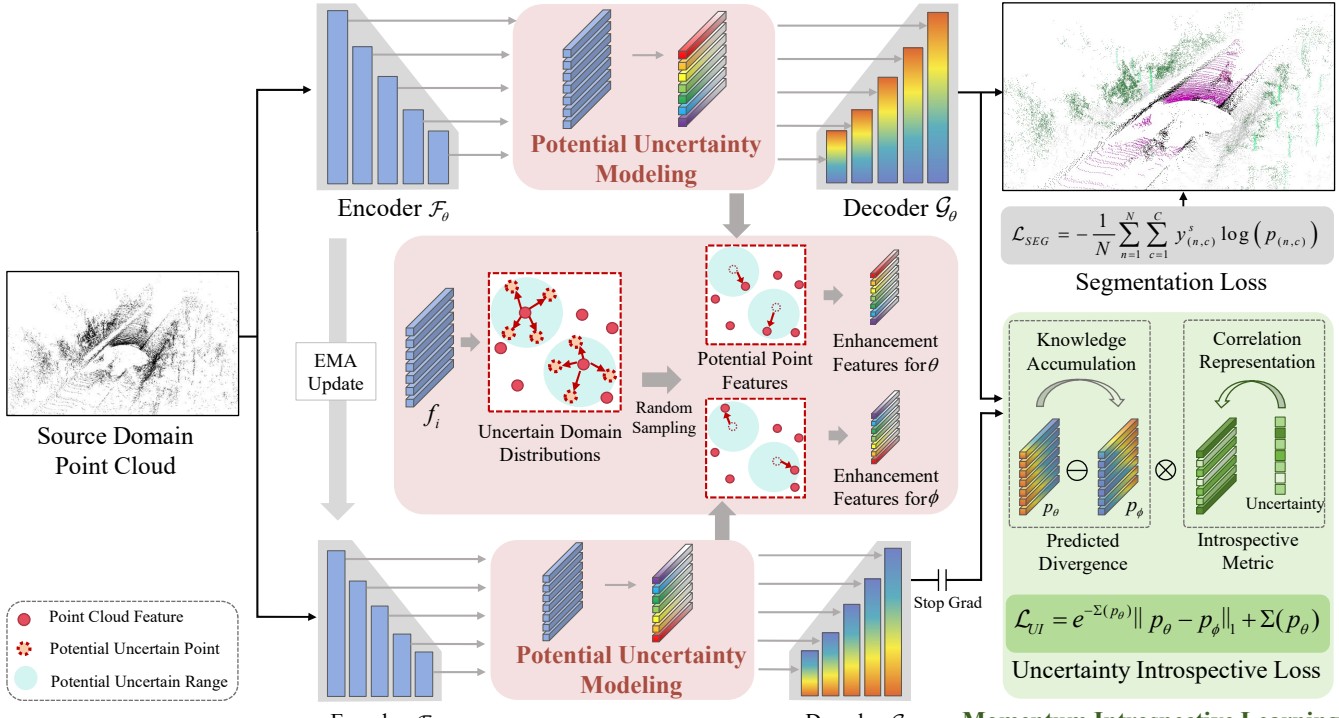

**Figure 2: Overview of the uncertainty introspective learning framework for domain generalized 3D segmentation. The Potential Uncertainty Modeling (PUM) explores the underlying uncertain point cloud features and generates the different distributions for each point. Momentum Introspective Learning (MIL) learns generalized feature representation through uncertainty introspection loss. The exponential moving average (EMA) is utilized to update the momentum network.**

of uncertainty is ignored during training, this will make unknown target domain data more confusing to the model. Therefore, the proposed method strives to focus on the uncertainty problem to improve the generalization of 3D point cloud segmentation.

### 2.3 Uncertainty Estimate

Uncertainties in 3D point cloud models are mainly due to data noise, missingness, incompleteness, and the model's limited ability to model complex environments[25, 33]. Uncertainty estimation methods can be used to reduce noise interference [16, 42]. PointRas [41] improves predictions at multiple resolutions using uncertainty selection criteria. HPAL[33] proposes active learning based on hierarchical points to improve semi-supervised point cloud segmentation performance by using uncertainty estimation. 3DPC-CISS [36] propagates labels within local neighborhoods to eliminate noise in uncertain pseudo-labels, and improve class-incremental learning on 3D point clouds through uncertainty-aware pseudo-labels.

Different from previous methods, the proposed method explores the potential of uncertainty estimation in improving model domain generalization performance. This is challenging because the target domain is unknown. Therefore, we try to expand the distribution range of the simulated target domain as much as possible, which enhances point features differently within a reasonable range. In addition, momentum introspective learning can further explore the generalization characteristics under uncertain distributions.

## 3 UNCERTAINTY INTROSPECTIVE LEARNING

This section introduces the proposed uncertainty introspective learning to improve the generalization of point cloud segmentation domains. Section 3.1 explains the basic issues of domain generalization for point cloud semantic segmentation, and introduces the design concept and framework of the proposed method. Section 3.2 details the proposed Potential Uncertainty Modeling to improve the model representation capacity. Section 3.3 expresses the proposed Momentum Introspective Learning to learn generalization features from uncertain distributions.

### 3.1 Problem Definition and the Framework

Generalized point cloud segmentation models are expected to be effectively implemented in unseen scene distributions. Set $\mathcal{S} = \{(x_n^s, y_n^s)\}_{n=1}^N$ as the source domain point cloud data, where $N$ is the number of point clouds, $x_n^s \in \mathbb{R}^4$ is the point spatial location information of 3D coordinate and the properties of points, $y_n^s \in \{1, 2, ..., C\}$ is the corresponding semantic label and $C$ is the number of semantic categories. The target domain point clouds are $\mathcal{T} = \{x_n^t\}_{n=1}^{N'}$, where $N'$ is the number of unseen target point clouds. The goal of point cloud domain generalization is to learn a generalized mapping function $\theta : \mathbb{R}^4 \rightarrow \{1, 2, ..., C\}$ by using source domain

data. This function aims to accurately predict the semantic label of each point for an arbitrarily distributed target domain.

The 3D semantic segmentation usually includes an encoder $\mathcal{F}_\theta$ and a decoder $\mathcal{G}_\theta$. Thus the segmentation result can be expressed as $\boldsymbol{p} = \mathcal{G}_\theta(\mathcal{F}_\theta(x^s))$, where $p \in \mathbb{R}^N$. The segmentation loss $\mathcal{L}_{SEG}$ for the point cloud calculated by the cross-entropy loss:

$$\mathcal{L}_{SEG} = -\frac{1}{N} \sum_{n=1}^{N} \sum_{c=1}^{C} y_{(n,c)}^s \log\left(p_{(n,c)}\right). \tag{1}$$

In addition to fitting the segmentation loss of the source-only model, further strategies, such as feature augmentation, need to be designed to improve generalization capabilities.

The framework of the proposed method is shown in Figure 2, which focuses on point uncertainty to enhance the feature representation of the domain. The Potential Uncertainty Modeling (PUM) explores uncertain domain distribution for generating different point features. In addition, the model needs to learn generally applicable features in uncertain distributions. Therefore, Momentum Introspective Learning (MIL) utilizes the momentum network of encoder $\mathcal{F}_\phi$ and decoder $\mathcal{G}_\phi$ for predicted Divergence, which combines with the introspective metric to learn generalizing features. The following sections will introduce PUM and MIL in detail.

## 3.2 Potential Uncertainty Modeling

PUM models the underlying uncertain point cloud features and enhances the point feature with different potential distributions. Usually, processing point cloud data requires continuous fusion of underlying features to restore spatial structure information, which is usually implemented using U-Net-like architecture. Therefore, We model uncertainty for each feature level and fuse it into the encoded features, which allows the model to learn the underlying data distribution top-down. In addition, aligning statistics from the source domain to the target domain allows the model to perform knowledge transfer [28], since feature statistics represent the distribution pattern of point cloud features.

Therefore, PUM is designed to model potential uncertainty with the statistics, as shown in Figure 3. Assume $f_i \in \mathbb{R}^{N_i \times K_i}$ is the $i$-th feature layer, where $N_i$ is the sample number of the $i$-th layer and $K_i$ is the dimension of the layer. First, the uncertainty factor $\mathbb{U}_i^s$ is obtained by calculating the statistic $s_i$ of each feature layer. Secondly, the point distribution $\mathbb{O}_i^s$ varies within the uncertainty range $\mathbb{U}_i^s$ by randomly generated $\alpha_i$. Finally, $f_i'$ is generated through scale and shift by the random statistical factor $s_i^{\mathbb{U}}$, and concatenated with the $j$-th decoder feature $g_j$ to obtain $g_j'$ for the next decoder layer.

More specifically, the uncertainty values of statistics can be obtained by calculating the variance. And the statistical vectors of the $i$-th feature layer are the mean $\mu_i$ and variance $\Sigma_i$. Therefore, the uncertainty factors $\mathbb{U}_i^\mu$ and $\mathbb{U}_i^\Sigma$ can be estimated as:

$$\mathbb{U}_i^\mu = Var(\mu_i), \quad \mathbb{U}_i^\Sigma = Var(\Sigma_i). \tag{2}$$

Then, the point distributions $\mathbb{O}_i^\mu$ and $\mathbb{O}_i^\Sigma$ can vary randomly within the uncertainty factors $\mathbb{U}_i^\mu$ and $\mathbb{U}_i^\Sigma$, which can be expressed as:

$$\mathbb{O}_i^\mu = \alpha_i \otimes \mathbb{U}_i^\mu, \quad \mathbb{O}_i^\Sigma = \alpha_i \otimes \mathbb{U}_i^\Sigma, \tag{3}$$

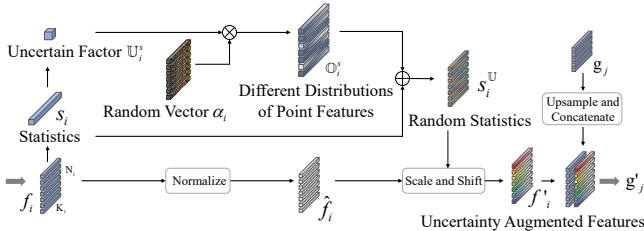

**Figure 3: Overview of the potential uncertainty modeling. The PUM is designed to generate the different potential distributions for each point and augment the point features.**

where $\alpha_i$ is a random vector and $\alpha_i \in \mathbb{R}^{N_i \times K_i}$. $\alpha_i$ takes a random value from the standard normal distribution during each forward propagation. Therefore, the different potential distributions for each point can be generated by random $\alpha_i$.

The random statistics $\mu_i^{\mathbb{U}}$ and $\Sigma_i^{\mathbb{U}}$ can be centered on the original statistics $\mu_i$ and $\Sigma_i$ of the feature. And take different values with the point distributions $\mathbb{O}_i^\mu$ and $\mathbb{O}_i^\Sigma$:

$$\mu_i^{\mathbb{U}} = \mu_i + \mathbb{O}_i^\mu, \quad \Sigma_i^{\mathbb{U}} = \Sigma_i + \mathbb{O}_i^\Sigma. \tag{4}$$

Then the uncertain augmentation feature $f_i'$ of the $i$-th layer can be generated by the random statistics $\mu_i^{\mathbb{U}}$ and $\Sigma_i^{\mathbb{U}}$:

$$f_i' = \Sigma_i^{\mathbb{U}} \hat{f}_i + \mu_i^{\mathbb{U}}, \tag{5}$$

where $\hat{f}_i$ is the normalized feature of $f_i$, which can be obtained by $\hat{f}_i = (f_i - \mu_i)/\Sigma_i$. Eq.5 enables the feature enhancement direction of $f_i'$ controlled within the adaptive range. Each point feature is enhanced to the different distribution and varies within the uncertainty range. This provides richer features than manually designed methods, which augment features in limited directions.

The uncertainty augmentation feature $f_i'$ of each layer needs to be integrated into the decoding features to learn invariance. Therefore, the decoder feature $g_j$ of $j$-th layer is upsampled to the same feature size as $f_i'$ and concatenated, then the formula of $g_j'$ is:

$$g_j' = Con(f_i', Up(g_j)). \tag{6}$$

In this way, PUM can model the uncertainty in the network from top to bottom, which learns the underlying distribution of the point cloud data. PUM supplements the training process with the underlying target domain distribution, enabling the model to adapt to various point cloud distribution information.

## 3.3 Momentum Introspective Learning

In order to introspectively learn generally applicable features from the uncertainty distribution, MIL needs to be designed to compare differences in uncertainty distributions and feedback to the model to carefully judge and understand the divergence. Specifically, the model utilizes the PUM module to obtain the results:

$$p_\theta = \mathcal{G}_\theta(PUM(\mathcal{F}_\theta(x^s))). \tag{7}$$

where $p_\theta \in \mathbb{R}^{N \times C}$ is the segmentation result based on uncertainty features. Assuming that the semantic difference is $D$, which can be intuitively measured using source domain labels $D(p_\theta, y^s)$.

However, the optimization of the segmentation loss already constrains $p_\theta$ to approximate $y^s$, and the optimization process cannot reflect the discrepancy in different distributions during the training process. Therefore, we utilize the knowledge of momentum accumulation to reflect the uncertainty difference.

Specifically, assuming that the update weight of the momentum network is $\delta$, the encoder $\mathcal{F}_\phi$ and decoder $\mathcal{G}_\phi$ can be expressed as:

$$\begin{aligned} \mathcal{F}_\phi &= \delta \mathcal{F}_\phi + (1 - \delta) \mathcal{F}_\theta, \\ \mathcal{G}_\phi &= \delta \mathcal{G}_\phi + (1 - \delta) \mathcal{G}_\theta. \end{aligned} \tag{8}$$

By inserting PUM into the momentum network, differential results of knowledge accumulation can be generated:

$$p_\phi = \mathcal{G}_\phi(PUM(\mathcal{F}_\phi(x^s))), \tag{9}$$

where $p_\phi$ has different potential distributions with $p_\theta$, which is helpful to compare the predicted divergence under uncertainty.

The divergence between $p_\phi$ and $p_\theta$ can be measured by:

$$D(p_\theta, p_\phi) = \|p_\theta - p_\phi\|_1, \tag{10}$$

where $\|\|_1$ denotes the L1-norm. A straightforward method is to reduce $D(p_\theta, p_\phi)$. However, the predicted divergence incorporates the underlying distribution established by PUM, and the uncertain differences should be carefully considered. Therefore, we used the prediction variance $\Sigma(p_\theta)$ to measure the predicted divergence. Besides, considering the semantic differences and ambiguity of point cloud results, when the uncertainty value of the model output is high, the model is expected to ignore the bias. That is, the model needs an introspective metric to carefully judge and understand the divergence.

Therefore, we validly map uncertainty into an introspective metric. First, the metric factor should be resolved as a negative correlation of uncertainty, with values in the valid positive domain. Therefore, $e^{-\Sigma(p_\theta)}$ is designed as the metric, thus avoiding a potential division by zero for stable training. Besides, it is worth noting that the model may predict larger variances to reduce the impact of differences. Therefore, $\Sigma(p_\theta)$ requires simultaneous regularization to optimize for introspective metrics. All in all, the model can be optimized with uncertainty introspective loss $\mathcal{L}_{UI}$, which can be expressed as:

$$\mathcal{L}_{UI} = e^{-\Sigma(p_\theta)} \|p_\theta - p_\phi\|_1 + \Sigma(p_\theta). \tag{11}$$

By minimizing $\mathcal{L}_{UI}$, the model selectively ignores the consistent bias when the uncertainty value of the model output is high, which avoids false pulls on ambiguous information. Therefore, assuming the loss balance parameter is $\lambda$, the overall training objective can be formulated as:

$$\min_\theta \mathcal{L}_{SEG} + \lambda \mathcal{L}_{UI}. \tag{12}$$

Training in this way can adapt to the underlying uncertainty distribution through PUM, and learn generalization features through MIL. Therefore, the inference process can directly use $\mathcal{F}_\theta$ and $\mathcal{G}_\theta$ to generate prediction results for the target domain.

## 4 EXPERIMENTS AND ANALYSIS

### 4.1 Datasets and Implement Details

*4.1.1 Data Set Description.* We utilize three datasets based on the previous benchmarks: Normal weather to adverse weather and virtual to real-world data distribution. **SemanticKITTI** [1] is point cloud data collected by LiDAR sensors in urban scenes under normal weather conditions. We use the training split with 19 point-level semantic category annotations as source data. **SynLiDAR** [30] is a synthetic LiDAR dataset collected from multiple virtual environments. The dataset is rich in scenes and layouts, which consists of more than 19 billion points. We also select the training set labeled with 19 categories as the source data. **SemanticSTF** [31] is a point cloud dataset of urban scenes collected under adverse weather conditions. Extreme weather includes fog, snow, and rain. We utilize SemanticSTF as the target domain.

*4.1.2 Implementation Details.* The proposed method is implemented on the PyTorch [20] platform. Following the previous work [31], we use the widely used MinkowskiNet [3] with sparse convolution [26] as the backbone for all models. The stochastic gradient descent (SGD) with a momentum of 0.9 is used as the optimizer to train the model. The batch size is set to 4, and the initial learning rate is 0.24 with a decay factor of 0.0001. The update parameter $\delta$ of the momentum network in Eq.8 is set to 0.999. The balance parameter $\lambda$ of Eq.12 is set to 3e-4, which is set according to the experimental results in Table 7. For the training processing of the source domain data, we use random data enhancement such as rotation from $[-\pi, \pi]$, scaling from $[0.95, 1.05]$, dropout with 0.2 rates, flipping, noise, and jitter to prevent overfitting.

### 4.2 Compared with the State-of-the-arts

The proposed method is compared with state-of-the-art methods on generalized point cloud benchmarks. Table 1 is the quantitative comparison on SemanticKITTI → SemanticSTF benchmark. Our method outperforms the current state-of-the-art domain generalization point cloud segmentation method by 6.9% in mIou, reaching 35.5%. PolarMix [29] mixes cross-scans of point clouds for data enhancement. PCL [37] uses a proxy to align positive samples from different domains. Maximum mean difference (MMD) is used in [13] to align distributions between different domains. PointDR [31] utilizes domain randomization technology for data enhancement. These methods enrich the distribution of source domain point clouds with manually designed feature enhancements. Our method performs better because we control the feature enhancement direction within an adaptive range, which has richer features than simply using a manually designed data enhancement method. Furthermore, the proposed method learns generalized features from the uncertainty in point cloud data, which facilitates generalization to unseen target domains.

Table 2 is the quantitative comparison on SemanticKITTI → SemanticSTF benchmark. SynLiDAR is a generated virtual street view data, which causes distribution differences with the real scene. Therefore, the generalization effect of the methods is slightly worse than the methods trained with the real-world scene source domain. Even so, our method exceeds the current state-of-the-art methods, achieving a mIoU of 21.1%. This is because the proposed method

**Table 1: Comparison of domain generalization on SemanticKITTI → SemanticSTF.**

| Model | car | bi.cle | mt.cle | truck | oth-v. | pers. | bi.clst | mt.clst | road | parki. | sidew. | oth-g. | build. | fence | veget. | trunk | terra. | pole | traf. | mIou |
|---|---|---|---|---|---|---|---|---|---|---|---|---|---|---|---|---|---|---|---|---|
| Noise-aug [31] | 74.4 | 0.0 | 0.0 | 23.3 | 0.6 | 19.7 | 0.0 | 0.0 | 60.3 | 10.8 | 33.9 | 0.7 | 72.0 | 45.2 | 58.7 | 17.5 | 42.4 | 22.1 | 9.7 | 25.9 |
| PolarMix [29] | 57.8 | 1.8 | 3.8 | 16.7 | 3.7 | 26.5 | 0.0 | 2.0 | 65.7 | 2.9 | 32.5 | 0.3 | 71.0 | 48.7 | 53.8 | 20.5 | 45.4 | 25.9 | 15.8 | 26.0 |
| PCL [37] | 65.9 | 0.0 | 0.0 | 17.7 | 0.4 | 8.4 | 0.0 | 0.0 | 59.6 | 12.0 | 35.0 | 1.6 | **74.0** | 47.5 | 60.7 | 15.8 | 48.9 | 26.1 | 27.5 | 26.4 |
| MMD [13] | 63.6 | 0.0 | 2.6 | 0.1 | **11.4** | 28.1 | 0.0 | 0.0 | 67.0 | 14.1 | 37.9 | 0.3 | 67.3 | 41.2 | 57.1 | 27.4 | 47.9 | 28.2 | 16.2 | 26.9 |
| PointDR [31] | 67.3 | 0.0 | 4.5 | 19.6 | 9.0 | 18.8 | 2.7 | 0.0 | 62.6 | 12.9 | 38.1 | 0.6 | 73.3 | 43.8 | 56.4 | **32.2** | 45.7 | 28.7 | **27.4** | 28.6 |
| Ours | **77.9** | **10.6** | **19.1** | **26.0** | 9.7 | **46.3** | **6.0** | **9.3** | **69.1** | **18.0** | **38.6** | **9.4** | 73.3 | **51.2** | **60.8** | 30.9 | **50.8** | **31.8** | 22.3 | **35.5** |

**Table 2: Comparison of domain generalization on SynLiDAR → SemanticSTF.**

| Model | car | bi.cle | mt.cle | truck | oth-v. | pers. | bi.clst | mt.clst | road | parki. | sidew. | oth-g. | build. | fence | veget. | trunk | terra. | pole | traf. | mIou |
|---|---|---|---|---|---|---|---|---|---|---|---|---|---|---|---|---|---|---|---|---|
| MMD [13] | 25.5 | 2.3 | 2.1 | 13.2 | 0.7 | 22.1 | 1.4 | **7.5** | 30.8 | 0.4 | 17.6 | 0.2 | 30.9 | 19.7 | 37.6 | 19.3 | 43.5 | 9.9 | 2.6 | 15.1 |
| Noise-Aug [31] | 27.1 | 2.3 | 2.3 | 16.0 | 0.1 | 23.7 | 1.2 | 4.0 | 27.0 | 3.6 | 16.2 | 0.8 | 29.2 | 16.7 | 35.3 | 22.7 | 38.3 | 17.9 | 5.1 | 15.2 |
| PCL [37] | 30.9 | 0.8 | 1.4 | 10.0 | 0.4 | 23.3 | 4.0 | 7.9 | 28.5 | 1.3 | 17.7 | **1.2** | 39.4 | 18.5 | 40.0 | 16.0 | 38.6 | 12.1 | 2.3 | 15.5 |
| PolarMix [29] | 39.2 | 1.1 | 1.2 | 8.3 | 1.5 | 17.8 | 0.8 | 0.7 | 23.3 | 1.3 | 17.5 | 0.4 | 45.2 | 24.8 | 46.2 | 20.1 | 38.7 | 7.6 | 1.9 | 15.7 |
| PointDR [31] | 37.8 | 2.5 | 2.4 | **23.6** | 0.1 | 26.3 | 2.2 | 3.3 | 27.9 | 7.7 | 17.5 | 0.5 | 47.6 | 25.3 | 45.7 | 21.0 | 37.5 | 17.9 | 5.5 | 18.5 |
| Ours | **43.3** | **2.8** | **2.6** | 23.2 | **3.2** | **31.3** | **2.5** | 4.4 | **34.3** | **9.2** | **17.9** | 0.3 | **57.1** | **27.6** | **50.0** | **24.2** | 41.5 | **19.0** | **6.1** | **21.1** |

**Table 3: Comparison of the SemanticKITTI → {Dense fog, Light fog, Rain, Snow}, which is domain generalization from normal to adverse weather in real street scenes.**

| Method | D-fog | L-fog | Rain | Snow | Mean |
|---|---|---|---|---|---|
| PolarMix [29] | 29.7 | 25.0 | 28.6 | 25.6 | 27.2 |
| Noise-Aug [31] | 29.3 | 25.6 | 29.4 | 24.8 | 27.3 |
| PCL[37] | 28.9 | 27.6 | 30.1 | 24.6 | 27.8 |
| MMD [13] | 30.4 | 28.1 | 32.8 | 25.2 | 29.1 |
| PointDR [31] | 31.3 | 29.7 | 31.9 | 26.2 | 29.8 |
| Ours | **36.3** | **34.5** | **35.5** | **33.3** | **34.8** |

can adaptively model uncertainty based on data characteristics and learn generalization features in uncertain-aware comparisons.

Table 3 is a quantitative comparison of the SemanticKITTI → {dense fog, light fog, rain, snow} benchmark, which verifies the generalization of the proposed method under different severe weather conditions. The proposed method outperforms the current state-of-the-art algorithms in all four weather conditions. This shows that adverse weather conditions bring uncertainty to the recognition. The proposed method can effectively complement the feature distribution under different conditions.

In addition, we visualized the point cloud segmentation results under different weather conditions, as shown in Figure 4. It can be seen that the segmentation results of our method are more similar to ground truth, which can further demonstrate the effectiveness of the proposed method through qualitative comparative experiments.

## 4.3 Ablation study and Analysis

In order to verify the effectiveness of the proposed modules, ablation experiments were conducted on the SemanticKITTI → Semantic-STF benchmark, as shown in Table 4.

The model trained with only cross-entropy loss is set as the baseline, and we supplemented the PUM and MIL modules, respectively, to prove the impact of the modules. The results show that adding PUM alone can greatly improve the baseline method by 2.7%, reaching a mIoU of 34.1%. This is because uncertainty in the data limits the model from learning the generalization distribution. PUM improves model representation by modeling the underlying uncertainty of features at each layer.

Adding MIL alone can increase mIoU by 0.9%, and mIoU reaches 32.3%. This is because the results generated by the momentum network can use accumulated knowledge to guide the current results to a certain extent. Adding MIL under the uncertainty distribution established by PUM can increase mIoU by 1.4% and reach 35.5%.

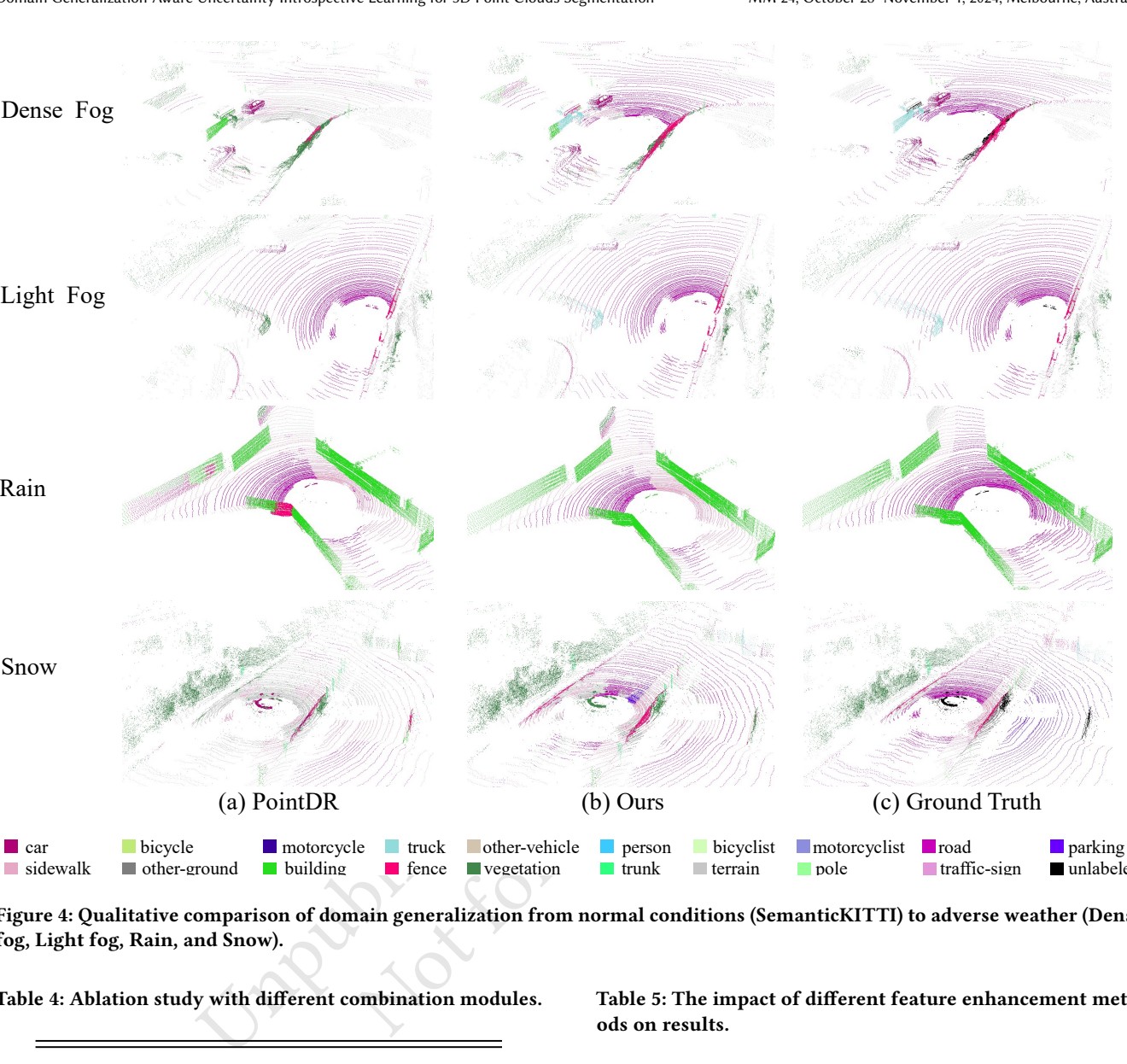

Dense Fog

Light Fog

Rain

Snow

(a) PointDR          (b) Ours          (c) Ground Truth

■ car    ■ bicycle    ■ motorcycle    ■ truck    ■ other-vehicle    ■ person    ■ bicyclist    ■ motorcyclist    ■ road    ■ parking
■ sidewalk    ■ other-ground    ■ building    ■ fence    ■ vegetation    ■ trunk    ■ terrain    ■ pole    ■ traffic-sign    ■ unlabeled

**Figure 4: Qualitative comparison of domain generalization from normal conditions (SemanticKITTI) to adverse weather (Dense fog, Light fog, Rain, and Snow).**

**Table 4: Ablation study with different combination modules.**

| Method | PUM | MIL | mIoU |
|--------|-----|-----|------|
| Baseline | - | - | 31.4 |
| Baseline + PUM | ✓ | - | 34.1 |
| Baseline + MIL | - | ✓ | 32.3 |
| Baseline + PUM + MIL | ✓ | ✓ | 35.5 |

This shows that the proposed MIL can allow the model to perform introspective learning in the uncertainty distribution, which is beneficial to further improving the feature generalization ability.

*4.3.1 The impact of Uncertainty Perception.* In order to further verify the impact of uncertainty on domain generalization of point cloud segmentation, we compared other feature augmentation

**Table 5: The impact of different feature enhancement methods on results.**

| Method | DataAug | Dropout | Uncertainty |
|--------|---------|---------|-------------|
| mIoU | 32.8 | 29.8 | 35.5 |

methods under the same settings and framework in Table 5. DataAug means that two networks learn different feature representations under data augmentation. In addition, we also compare the method that utilizes dropout layers to generate different results with Bayesian posterior probabilities. Experimental results show that the proposed uncertainty-aware method has better performance. This is because the feature enhancement direction is controlled within the adaptive range, which can produce richer features than simply using manually designed data enhancement methods.

**Table 6: The impact of adding PUM on different levels of feature layers.**

|  | L.5 | L.4 | L.3 | L.2 | L.1 | mIoU |
|---|---|---|---|---|---|---|
| (a) | - | - | - | - | - | 32.3 |
| (b) | ✓ | - | - | - | - | 33.1 |
| (c) | ✓ | ✓ | - | - | - | 34.3 |
| (d) | ✓ | ✓ | ✓ | - | - | 34.5 |
| (e) | ✓ | ✓ | ✓ | ✓ | - | 35.5 |
| (f) | ✓ | ✓ | ✓ | ✓ | ✓ | 33.1 |

**Table 7: The influence of the parameter $\lambda$ of the uncertainty introspection loss on experimental results.**

| $\lambda$ | 5e-3 | 1e-3 | 5e-4 | 3e-4 | 1e-4 |
|---|---|---|---|---|---|
| mIoU | 31.6 | 34.8 | 35.1 | 35.5 | 33.5 |

*4.3.2 The level of Uncertainty Modeling.* Additional experiments further validate the impact of different uncertainty levels modeled in PUM, as shown in Table 6. L5 to L1 indicates modeling uncertainty on the feature layer of the corresponding level, whereas L5 means modeling uncertainty on the deepest semantic features. It can be seen that modeling uncertainty on relatively deep feature layers is beneficial to improving accuracy. This is because the uncertainty established by deep features can be effectively learned by subsequent feature layers. On the contrary, the uncertain modeling of the shallow feature L1 affects the further improvement of mIoU. This is because the L1 features are close to the output, and enhancement will interfere with the distribution of classification features. Experiments have proven that modeling uncertainty on L5 to L2 can achieve the best performance.

*4.3.3 The Effect of Momentum Introspective Learning.* The impact of MIL on the results can be tested by adjusting the weight of the uncertainty introspective loss, as shown in Table 7. The value of $\lambda$ between [5e-3,1e-4] has large fluctuations in the results. It can be seen from the experiment that excessive weight makes the model excessively introspective, which is not conducive to the model itself fitting the data distribution. Experiments show that when the parameter is 3e-4, the model can achieve the best performance.

*4.3.4 The Effect of Different Distance Measures.* Table 8 compares the impact of different distance measurement methods on the results. Manhattan Distance intuitively considers the distance between two variables. Therefore, when there is great uncertainty in the prediction results, the Manhattan distance is more robust than the Euclidean distance and is less susceptible to outliers. Additionally, cosine distance measures the direction of a variable, which means that differences in values are not fully taken into account.

**Table 8: The impact of different distance measures on experimental results.**

| $D(p_\theta, p_\phi)$ | Formula | mIoU |
|---|---|---|
| *Manhatton Distance* | $\|p_\theta - p_\phi\|_1$ | 35.5 |
| *Euclidean Distance* | $\sqrt{(p_\theta - p_\phi)^2}$ | 34.6 |
| *Cosine Distance* | $\frac{p_\theta \, p_\phi}{\sqrt{p_\theta^2} \times \sqrt{p_\phi^2}}$ | 34.7 |

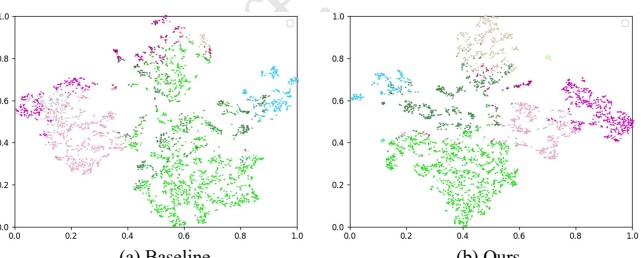

**Figure 5: Features visualized by t-SNE. Clusters of different colors represent projections of different classes.**

Therefore, the Manhattan distance measurement method is more suitable for the proposed method.

*4.3.5 Features Visualized.* Figure 5 provides the qualitative analysis of features via t-SNE [27]. We compared feature embeddings before and after uncertainty perception. It can be seen that the proposed method is more cautious in feature embedding than the baseline method. Our method slightly pulls clusters of the same category together, which can prevent the impact of point uncertainty domain shift.

## 5 CONCLUSIONS AND DISCUSSION

In this paper, a Domain Generalization-Aware Uncertainty Introspection Learning (DGUIL) method is proposed for 3D point cloud segmentation, which addresses the point uncertainty in domain shift to improve feature generalization. Potential Uncertainty Modeling (PUM) and Momentum Introspective Learning (MIL) are proposed to model and learn the uncertainty distributions. PUM explores the underlying uncertain point cloud features and enhances the point feature with different potential distributions. MIL learns generalization features from uncertain distributions with uncertainty introspection loss. Experiments demonstrate the advantages of our method. Furthermore, the point cloud is the collection from real-world 3D space, and its distribution conforms to objective physical phenomena. Future work can incorporate physical priors to learn point cloud geometric distribution and use interpretable theorems to improve generalization.

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
