# OpenReview forum: "Domain Generalization-Aware Uncertainty Introspective Learning for 3D Point Clouds Segmentation"
_acmmm.org/ACMMM/2024/Conference — MM2024 Poster_

### Official Review · Reviewer_67kX · 2024-05-12

**Rating:** 5
**Confidence:** 2

**Summary:**

A Domain Generalization-Aware Uncertainty Introspection Learning (DGUIL) method is proposed for 3D point cloud segmentation, which addresses the point uncertainty in domain shift to improve feature generalization.

**Strengths:**

- The performance is good.

- This paper proposes a new insight by generating different potential distributions for each point and augmenting the point features.

**Limitations:**

## Main weakness

1. In Tabel 1-3, add more baseline methods, like references[1][2]

2. Why the performance of the baseline in Table 4  is better than that of other methods in Table 1? Maybe it is unfair. The most contribution of performance is from the baseline in Table 4.



## Other weakness

1. In line 342, what is the properties of points?

2. Tense inconsistency. In line 490， "used" -> "use". In line 672， "visualized" -> "visualize"

3. In line 573, "SemanticKITTI → SemanticSTF" -> "SynLiDAR → SemanticSTF"

4. Tabel 1-2 are over the line.

5. circle the key parts in Figure 4

6. highlight  your experiment configs in ablation studies



## Reference

[1] Kim H, Kang Y, Oh C, et al. Single domain generalization for lidar semantic segmentation[C]//Proceedings of the IEEE/CVF Conference on Computer Vision and Pattern Recognition. 2023: 17587-17598.

[2] Sanchez J, Deschaud J E, Goulette F. Domain generalization of 3D semantic segmentation in autonomous driving[C]//Proceedings of the IEEE/CVF International Conference on Computer Vision. 2023: 18077-18087.

**Suitability:**

2

---

### Official Review · Reviewer_HC3Z · 2024-05-19

**Rating:** 5
**Confidence:** 3

**Summary:**

This paper studies domain generalization on the point cloud data. In addition, a method, termed Domain Generalization-Aware Uncertainty Introspective Learning (DGUIL) is proposed. Specifically, uncertainty metrics and momentum introspective learning are adapted. Extensive experiments show the effectiveness of the proposed method.

**Strengths:**

+ The proposed method is interesting.
+ The experimental results are impressive.
+ This paper is well originalized.

**Limitations:**

- In this work, the main experiments are conducted within MinkowskiNet, can the proposed method work well on another backbone network？
- The hyper-paramters δ ccan be studied.
- Can the visualization results on the manual selection-based feature augmentation provided?

**Suitability:**

3

---

### Official Review · Reviewer_x2sV · 2024-05-25

**Rating:** 4
**Confidence:** 3

**Summary:**

This paper addresses the challenge of domain generalization in 3D segmentation, focusing on learning from point clouds with unknown distributions. The study first identifies a common issue in previous work: the limited scope of manually designed feature augmentation. To overcome this, this work proposes a framework that automatically generates and learns from diverse distributions for each point, based on the underlying uncertain features of the point clouds.

**Strengths:**

- The methodology is well-motivated. Previous works primarily focus on manually designing feature augmentations for domain generalization in 3D segmentation. In contrast, the proposed DGUIL emphasizes the uncertainty of points, introducing 1) a Point Uncertainty Module (PUM) to explore uncertainty features and generate distinct distributions for each point, and 2) a Multi-Instance Learning (MIL) approach to learn generalized feature representations in uncertain distributions. These modules are well-integrated and effectively address issues identified in prior studies.
- The DGUIL demonstrates promising performance on selected datasets.
- The visual aids are excellent. Figures 1 and 2 are clear, aesthetically pleasing, and facilitate the understanding of the framework. Figure 4 effectively visualizes the segmentation predictions intuitively.
- The writing is clear, well-structured, easy to follow, and meets the standards of a high-quality academic paper.

**Limitations:**

- The current work lacks a review of related literature and requires additional experiments for a thorough comparison [1][2]. While the performance on SemanticSTF is commendable, prior studies [1][2] have conducted extensive experiments on diverse datasets such as nuScenes-lidarseg [4], Waymo [5], SemanticPOSS [6], SemanticKITTI32 [3], Panda64 [7], and PandaFF [7]. These studies also applied the DG approach to various backbones including KPConv [8], SPVCNN [9], C3D [10] and so on. To validate the proposed DGUIL method, it is essential to evaluate it on multiple datasets and backbones. Without this, there remains scepticism about the method's applicability beyond a limited dataset or backbone.

- There is a minor typo in line 461: NxC should be in the index notation of R.

[1] Sanchez, J., Deschaud, J. E., & Goulette, F. (2023). Domain generalization of 3D semantic segmentation in autonomous driving. In Proceedings of the IEEE/CVF International Conference on Computer Vision (pp. 18077-18087).

[2] Kim, H., Kang, Y., Oh, C., & Yoon, K. J. (2023). Single domain generalization for lidar semantic segmentation. In Proceedings of the IEEE/CVF Conference on Computer Vision and Pattern Recognition (pp. 17587-17598).

[3] Behley, J., Garbade, M., Milioto, A., Quenzel, J., Behnke, S., Stachniss, C., & Gall, J. (2019). Semantickitti: A dataset for semantic scene understanding of lidar sequences. In Proceedings of the IEEE/CVF international conference on computer vision (pp. 9297-9307).

[4] Caesar, H., Bankiti, V., Lang, A. H., Vora, S., Liong, V. E., Xu, Q., ... & Beijbom, O. (2020). nuscenes: A multimodal dataset for autonomous driving. In Proceedings of the IEEE/CVF conference on computer vision and pattern recognition (pp. 11621-11631).

[5] Sun, P., Kretzschmar, H., Dotiwalla, X., Chouard, A., Patnaik, V., Tsui, P., ... & Anguelov, D. (2020). Scalability in perception for autonomous driving: Waymo open dataset. In Proceedings of the IEEE/CVF conference on computer vision and pattern recognition (pp. 2446-2454).

[6] Pan, Y., Gao, B., Mei, J., Geng, S., Li, C., & Zhao, H. (2020, October). Semanticposs: A point cloud dataset with large quantity of dynamic instances. In 2020 IEEE Intelligent Vehicles Symposium (IV) (pp. 687-693). IEEE.

[7] P. Xiao et al., "PandaSet: Advanced Sensor Suite Dataset for Autonomous Driving," 2021 IEEE International Intelligent Transportation Systems Conference (ITSC), Indianapolis, IN, USA, 2021, pp. 3095-3101, doi: 10.1109/ITSC48978.2021.9565009.

[8] Thomas, H., Qi, C. R., Deschaud, J. E., Marcotegui, B., Goulette, F., & Guibas, L. J. (2019). Kpconv: Flexible and deformable convolution for point clouds. In Proceedings of the IEEE/CVF international conference on computer vision (pp. 6411-6420).

[9] Tang, H., Liu, Z., Zhao, S., Lin, Y., Lin, J., Wang, H., & Han, S. (2020, August). Searching efficient 3d architectures with sparse point-voxel convolution. In European conference on computer vision (pp. 685-702). Cham: Springer International Publishing.

[10] Zhu, X., Zhou, H., Wang, T., Hong, F., Ma, Y., Li, W., ... & Lin, D. (2021). Cylindrical and asymmetrical 3d convolution networks for lidar segmentation. In Proceedings of the IEEE/CVF conference on computer vision and pattern recognition (pp. 9939-9948).

**Clarify of the Initial Rating:**

This paper is generally of good quality in terms of writing, presentation, motivation, methods, and results. However, there are some missing related works, and additional experiments on more datasets and backbones are needed to further validate the proposed method. This is an initial rating and it may be adjusted based on whether these concerns are addressed.

**Suitability:**

2

---

### Meta-Review · Area_Chair_ZzEZ · 2024-07-13

**Recommendation:** Accept (Poster)
**Confidence:** 4

**Metareview:**

This work proposes a framework that automatically generates and learns from diverse distributions for each point. It is based on the uncertain features of the underlying point clouds. This paper is well-written and presents strong motivation, methods, and results. Additionally, all the reviewers have given positive scores for this paper. Therefore, I recommend accepting this paper and suggest that the authors include the content discussed during rebuttal in the revised version.